# The Mental Lexicon Features of the Hakka-Mandarin Dialect Bilingual

**DOI:** 10.3390/brainsci12121629

**Published:** 2022-11-28

**Authors:** Yao Chen, Rong Zhou

**Affiliations:** 1School of Foreign Studies, South China Normal University, 55 West Zhongshan Ave, Tianhe District, Guangzhou 510631, China; 2The Center for Language Cognition and Assessment, School of Foreign Studies, South China Normal University, 55 West Zhongshan Ave, Tianhe District, Guangzhou 510631, China

**Keywords:** dialect bilingual speakers, mental lexicon, complex-network analysis, semantic fluency task, cross-language long-term repetition priming

## Abstract

The current study investigated the mental lexicon features of the Hakka-Mandarin dialect bilingual from two perspectives: the structural features of lexicons and the relations between lexicons. Experiment one used a semantic fluency task and complex-network analysis to observe the structural features of lexicons. Experiment two used a cross-language long-term repetition priming paradigm to explore the relations between lexicons, with three sub-experiments focusing on conceptual representation, lexical representation, and their relations, respectively. The results from experiment one showed that the dialect bilingual lexicons were small-world in nature, and the D2 (Mandarin) lexicon was better organized than the D1 (Hakka) lexicon. Experiment two found that D1 and D2 might have partially shared conceptual representations, separate lexical form representations, and partially shared lemma representations. Based on the findings, we tentatively proposed a two-layer activation model to simulate the lexicon features of dialect bilingual speakers.

## 1. Introduction

The bilingual mental lexicon has been explored for decades, yielding fruitful results. However, the mental lexicon of dialect bilingual speakers, a variation of bilingual, has been examined very little. Dialect bilingual speakers refer to those who speak more than one dialect [1,2]. Among the dialects they speak, one is frequently used for social interactions with family, namely the first dialect (D1). The other one is often used in a formal setting, such as an educational setting, namely the second dialect (D2) [3].

Similar to the bilinguals who speak two distinct languages, the dialects spoken by dialect bilingual speakers may have distinct phonetic and syntactic systems [4]. However, bilinguals and dialect bilingual speakers differ in terms of mutual intelligibility and political and social conventions [1]. In terms of the intelligibility criterion, the languages spoken by bilinguals are not mutually intelligible, while the dialects spoken by dialect bilingual speakers are mutually intelligible [1]. Nevertheless, there are dialects that are not mutually intelligible, such as Hakka and Mandarin in Chinese dialects, and the criterion may not count. In this case, political and social conventions may serve as a more suitable criterion. Even though the dialects are not mutually intelligible, they both belong to the variations of Chinese and henceforth are recognized as dialects [5]. Moreover, the dialects may share more linguistic similarities compared with the separate languages spoken by bilinguals. For example, Hakka and Mandarin share many cognates that originated from ancient Chinese [1].

The similarities and differences between bilinguals and dialect bilingual speakers cause us to wonder about the typical features dialect bilingual lexicons may demonstrate and whether the models on bilingual lexicons are suitable for simulating dialect bilingual lexicons. Hence, the current study intended to explore dialect bilingual lexicon features in the hope of deepening our understanding of dialect bilingual lexicon and gaining some insights for the construction of a more generalizable bilingual lexicon model which may incorporate dialect bilingual lexicon features into consideration.

### 1.1. The Structure of Bilingual Mental Lexicons

The bilingual mental lexicon has been explored from two perspectives: the structural features of lexicons and the relations between lexicons [6]. Concerning the structural features of lexicons, studies have focused on how words become organized and in what form lexicons exist. As for the relations between lexicons, studies have focused on the relations between the first language (L1) and the second language (L2) on the conceptual and lexical representations [7]. A conceptual representation refers to the semantic information of words, and a lexical representation refers to lexical information, including phonological, morphological, and syntactic information [7].

#### 1.1.1. Structural Features of Bilingual Lexicons

Models on the structural features of lexicons have regarded lexicons as a network where words are nodes, and nodes become connected with their relations with one another [8]. Among the relevant models, the revised spreading activation model (SAM) was the most prevalent, which assumed that words became connected with semantic similarities and syntactic, morphological, and phonological similarities [9].

The structural features of lexicons have been explored with semantic fluency tasks [10]. The task asks participants to produce as many words as possible within a semantic category in a limited time. The animal category is the most used because it is stable across languages and cultures, making results comparable [11]. Additionally, the color category is often used for further investigation because color bears relevance to the cultural background [12]. Such a category thus offers researchers an opportunity to observe a divergent domain from the animal category on the lexicon structure.

The data of a semantic fluency task were analyzed by clustering and switching, which was replaced by complex-network analysis. This is because the traditional approach involves subjective judgment, undermining the reliability of the outcome [10]. Whereas complex-network analysis can analyze the data with network science algorithms, rendering the results more objective and thus more reliable [10]. Complex-network analysis regards the systems in focus as networks where the nodes represent individual entities, and the edges represent the connections between entities [13]. It provides a tool for researchers to model a lexicon as a network graph and analyze the network characteristics [14]. With the merits of the approach, burgeoning research used it to investigate bilingual lexicon structures. Before reviewing the relevant research, we first introduced the statistical network features as follows.

Small-world property and community structure are the most explored network features [15]. A small-world property refers to a network with high local clustering and short global distance, where the activation could spread efficiently, and the structure of the robust system may not be easily influenced by outer interferences [15]. This property is characterized by the clustering coefficient (CC) and average shortest path length (ASPL) [15]. CC refers to the probability that any two nodes are neighbors whose values indicate the density of the neighborhoods within a network [16]. The average shortest path length (ASPL) refers to the mean of the steps along the shortest path for all pairs of nodes, whose values indicate the closeness among the nodes [16]. The small-world property of an empirical network could be identified when the CC and ASPL of the network are larger and smaller than the counterparts of a random network, respectively [17]. An empirical network refers to a system that is generated with empirical data, whereas a random network refers to a system that is generated with the same number of nodes as the empirical network and a fixed probability. Small-world properties could also be observed when the small-world index (S index) is larger than one [15]. Concerning community structure, it is often observed with the modularity index (Q index). The Q index measures how many communities that a network could be partitioned into, whose values indicate the modularity of a network [16].

Using complex-network analysis, studies on the bilingual lexicon found that both the L1 lexicon and the L2 lexicon were small-world in nature, in spite of the language differences [10,18]. It meant that words within the L1 lexicon and the L2 lexicon were closely connected, and activation within the systems could travel quickly. In addition, Wilks and Meara [18] found that, with sparser neighborhoods and fewer complex structures, the L2 lexicon was less well-organized than the L1 lexicon. Borodkin et al. [10] also found that, with denser neighborhoods and less community structure, the L2 lexicon was less well-organized than the L1 lexicon. The previous work attributed the less well-organized structure of the L2 lexicon to the less use frequency in the L2 lexicon. However, so far, the relevant studies on the bilingual lexicon are still limited, let alone the dialect bilingual lexicon. Whether the dialect bilingual lexicon displays a similar structure to its bilingual counterpart and which factors modulate the dialect bilingual lexicon await more research for further understanding.

#### 1.1.2. Relations between Bilingual Lexicons

Concerning the relations between lexicons, the revised hierarchical model (RHM) [19] believes that L1 and L2 have a shared conceptual representation and separate lexical representations, as displayed in panel A of Figure 1. The RHM assumes that at the beginning of L2 learning, the L1 lexicon could directly access a conceptual representation, while the L2 lexicon could only access a conceptual representation via the L1 lexical representation. This leads to an asymmetrical priming effect, with a larger effect in the L1-L2 priming condition than in the reverse direction. Nevertheless, as L2 proficiency developed, the L2 lexicon was associated more directly with a conceptual representation, leading to a symmetrical priming effect. The RHM was supported by numerous studies [20], but the shared conceptual representation was problematic because words within a translation pair might not have the same semantic features across languages and cultures [21]. If a conceptual representation were completely shared, a symmetrical priming effect would be obtained regardless of L2 proficiency rather than the asymmetrical priming effect proposed by the RHM [22].

To tackle this problem, the sense model proposed that L1 and L2 had a partially shared conceptual representation, as illustrated in panel B of Figure 1 [23]. The conceptual senses of a word included the overlapped senses and specific senses belonging either to L1 or L2. The sense model attributed the asymmetrical priming effect to language dominance, which incorporates language proficiency and language use frequency [24]. Specifically, if L1 were more dominant, the conceptual senses in L1 would be more prominent than in L2. Hence, in the L1-L2 priming direction, L1 words would activate most of the senses of L2 words, producing a priming effect. While in the reverse direction, L2 words could only activate the partial senses of L1 words, leading to an absence of the priming effect. If L2 was as prominent as L1, then the priming effect in both directions would be obtained. However, the model also has a limitation: it has a rough discussion on lexical representation.

The separate lexical representations proposed by the model mentioned above neglected that L1 and L2 shared cognates in some cases, which was addressed by the WEAVER++ model [25,26,27]. As shown in panel C of Figure 1, to better simulate the cognates within lexicons, WEAVER++ added a lemma representation that stored the syntactic information of words. The lemma information could serve as the prototype for the production of lexical forms. For cognates, the shared lemma would activate different lexical forms. Additionally, WEAVER++ specifies the connections between representations to simulate the mechanisms of lexical perception and lexical production. According to panel C of Figure 1, during lexical perception, the lexical form representation would first become activated. Then, the activation would flow to the lemma representation, which finally accessed the conceptual representation to retrieve the meaning of a word. During lexical production, the conceptual representation would first be activated. The activation then would transfer to the lemma representation, after which the lexical form representation would become activated to retrieve the word forms needed for lexical production, including the morphological and phonological forms. Nevertheless, the model did not provide a comprehensive discussion on the conceptual representation and the interaction between L1 and L2 in the conceptual and lexical representations.

To examine bilingual models, the cross-language long-term repetition priming paradigm has often been used [2,28,29]. Based on implicit memory, the paradigm, comprising a study phase and test phase, could rid the results of the influence of translation and memory strategies [29]. The study phase involves information encoding, and the test phase involves information extraction, and a cross-language long-term priming effect would be obtained if the encoding process recurs in the extraction process [28]. In both phases, the participants would be presented with learned words (which appear in the study phase) and unlearned words, during which they need to finish the animacy decision task or lexical decision task. The animacy decision task taps into the conceptual representation, and the lexical decision task draws upon the lexical representation, which is because the former task becomes more involved with conceptual processing while the latter task is more associated with perceptual processing [29]. This means that we could investigate the relations between lexicons from different perspectives using different tasks in different phases.

In summary, fruitful findings were obtained on the bilingual lexicon structure. However, few attempts were conducted in exploring the dialect bilingual lexicon structure.

### 1.2. Dialect Bilingual Mental Lexicon Structure

Concerning the structural features of the dialect bilingual lexicon, few studies exist. Compared with the bilingual lexicon, whether the dialect bilingual lexicon displays a similar or different structure and how words within the lexicon become organized remain unsolved questions.

As for the relations between lexicons, the existing studies mainly focused on conceptual representation [2,30]. Zhang and Zhang [30] used a cross-language long-term repetition priming paradigm to investigate the conceptual representation of Cantonese-Mandarin dialect bilingual speakers. They merely obtained a D1-D2 priming effect. Yi et al. [2] employed a repetition priming paradigm to explore the conceptual representation of Mandarin-Cantonese and Cantonese-Mandarin dialect bilingual speakers. Similarly, Yi et al. merely gained a priming effect in the D1-D2 direction. The absence of a symmetrical priming effect despite the equally high proficiencies in the two dialects of these dialect bilingual speakers indicates that D1 may connect more tightly with the conceptual representation regardless of language proficiency [2]. In this case, the age of acquisition of dialects may be a more influential factor in the configuration of a conceptual representation [31]. Since participants in the previous work acquired D1 at an earlier age than D2, the conceptual senses of D1 may be more entrenched than those of D2, resulting in a tighter connection between D1 and a conceptual representation. Moreover, the existing studies only focused on the conceptual representation while ignoring the lexical representation, and this creates difficulty for us to understand the relations between dialects comprehensively [32]. Furthermore, the previous work only focused on the type of dialect bilingual speakers whose dialects were very phonologically distinct and lexically similar to Cantonese and Mandarin, neglecting other types of dialects that share more linguistic similarities. Researching this type of dialect bilingual speaker may shed light on understanding the dialect bilingual mental lexicon.

A Hakka-Mandarin dialect bilingual speaker in the Guangdong province of China is a typical example of this type of speaker. According to the research findings on the evolution and linguistic distance of Chinese dialects, Hakka was found to be an admixture dialect of northern Chinese dialects, such as Mandarin, and southern dialects, such as Cantonese [33,34]. Similar to Mandarin, Hakka also originated from ancient Chinese and used the Han writing system as orthography [35]. This confers more linguistic similarities between Hakka and Mandarin, like a large proportion of cognates [1,33], in contrast to more apparent differences between Cantonese and Mandarin that were the focus of previous work. Given the special characteristics of such a type of speaker, the lexicon structure of Hakka-Mandarin dialect bilingual speakers “deserves due attention from Chinese academia” [36].

Henceforth, the current study intends to examine the mental lexicon features of Hakka-Mandarin dialect bilingual speakers, focusing on two questions: (1) what are the structural features of Hakka-Mandarin dialect bilingual lexicons? To be specific, are D1 and D2 lexicons small-world in nature? How do the words within the D1 and D2 lexicons become organized? (2) What are the relations between the lexicons? This second question includes three sub-questions: What are the relations between D1 and D2 in conceptual representation? What are the relations between D1 and D2 in lexical representation? Furthermore, what are the relations between the conceptual and lexical representations? By investigating these questions, we could observe the dialect bilingual lexicon features more comprehensively. Moreover, we could also gain more insights into the development of the theory on the bilingual mental lexicon.

## 2. Experiment One

To answer the first question, we used a semantic fluency task and complex-network analysis because, as mentioned above, the semantic fluency task is commonly used in exploring the issue, and complex-network analysis renders the results more objective.

### 2.1. Method

#### 2.1.1. Participants

Thirty-two students (nine males) from a university participated in the experiment and received a small amount of money for their participation. Before the experiment, they completed a modified bilingual language profile questionnaire [37], which required them to report their age of acquisition of the dialects and self-rate scores for their proficiency in D1 and D2 in terms of listening and speaking competencies (Hakka native speakers seldom use Hakka in reading or writing). The results showed that the participants were exposed to Hakka since birth and began learning Mandarin around the age of 4.8. Their proficiency in listening (*M*-Hakka = 4.91, *M*-Mandarin = 5.38, *t* = 1.935, *p* > 0.05) and speaking (*M*-Hakka = 5.03, *M*-Mandarin = 4.56, *t* = −1.754, *p* > 0.05) did not differ significantly between D1 and D2. Additionally, according to their verbal report, the participants used D1 more frequently during their interaction with family, while they used D2 more frequently in a formal setting. As college students, they had been immersed in an educational setting, leading to their higher use frequencies in D2.

#### 2.1.2. Design

We used a semantic fluency task with animal and color as semantic categories. As mentioned above, the animal category embodies more general world knowledge, and the color category represents more of the cultural background [12]. Moreover, in both D1 and D2, there is a large number of words in the animal and color domains because these words are essential in the daily life of the speakers. Researching the lexicon structure with these categories offered us a more representative picture of the structural features of the dialect bilingual lexicon.

#### 2.1.3. Procedure

Before the experiment, the participants practiced the task with family and furniture as semantic categories. Then they began using D1 or D2 to produce the words belonging to either the animal or color category within one minute. The sequences of the dialects and the semantic categories were counterbalanced. The oral responses were recorded and transcribed into Excel form.

### 2.2. Data Processing

The data processing included data preprocessing, the construction of a word correlation lexical network, the calculation of network parameters and network visualization, and the validation of the network analysis.

During the data preprocessing, the Hakka-color, Hakka-animal, Mandarin-color, and Mandarin-animal lexical matrices were first collected. Subsequently, they were converted to incidence matrices, where each top-line listed all the words the whole sample produced, and the rows presented the value of the responses of the participants. If a participant produced a certain word, the value in the relevant cell would be 1. If not, the value would be 0. All data were analyzed with the igraph package (version 1.3.5, Csardi & Nepusz) [38] of R software (version 4.1.2, R core team, Vienna, Austria) [39].

As for the construction of the word correlation lexical network, we considered the words that the participants produced as the nodes and the correlation coefficients between the words as weights on the edges [13]. Weights, specifically, the correlation coefficients, were obtained by converting the four incidence matrices into correlation matrices. With the nodes and weights, four weighted and undirected networks were constructed. However, the correlation coefficients between the words were relatively low, leading to an absence of some topological relations. This was tackled using the minimal spanning tree, which is a weighted subgraph with minimum total edge weight. This approach is often used to maximize the relations between the nodes within a network and to delete the edges with relatively low weights [40].

Additionally, to observe the network features, we obtained network visualizations and network parameters, including the number of nodes, CC, ASPL, S index, and Q index. Moreover, to validate the network analysis, we constructed Erdös-Rényi random networks to see whether the empirical networks differed significantly from the random networks [41].

### 2.3. Results and Analysis

#### 2.3.1. Word Correlation Lexical Network and Network Visualization

The network parameters are displayed in Table 1, which shows that the CC and ASPL of all four lexical networks were larger and smaller than the corresponding values of the random networks, respectively. Moreover, the small-world indexes were all larger than 1. This suggests that all the lexical networks were small-world in nature [15].

Moreover, in the animal category, the CC and Q indexes of the D2 lexical networks were larger than the ones of the D1 networks (CC-Mandarin = 0.986; CC-Hakka = 0.979; Q-Mandarin = 0.782; Q-Hakka = 0.731), indicating that the D2 networks were more densely connected and more modular. In other words, the D2 lexicon displayed denser neighborhoods and more separations among communities, where the lexical nodes could connect with each other easily, and information could be transferred efficiently. For the color category, the CC and Q indexes of the D1 lexicon were larger and smaller than the counterparts of the D2 lexicon, respectively (CC-Mandarin = 0.970; CC-Hakka = 0.994; Q-Mandarin = 0.722; Q-Hakka = 0.720), indicating that the D1 networks were more densely connected and less modular. Namely, the words within the D1 lexicon displayed denser neighborhoods and a lower likelihood of grouping into subcategories. Consequently, words within the D1 lexicon might become trapped in rigid neighbors and thus render it difficult for the lexical information to transfer [10]. Taken together, the results from different categories both indicated that the lexical information within the D2 lexicon would become transferred in a more efficient manner.

Furthermore, to see how the words became organized, we obtained network visualizations, which are displayed in Figure 2, Figure 3, Figure 4 and Figure 5. The different colors in the figures represent the different clusters of lexical nodes. As shown in these figures, the words were organized in a random manner. However, some words tended to cluster according to semantic, morphological, syntactic, and phonological similarities. For example, in Figure 2, the nodes “chicken”, “goose”, and “fish” were associated because all these animals are domestic animals. In Figure 5, the nodes “dark red”, “scarlet”, and “rose red” were clustered, which might be because they all represent different degrees of the color red. Furthermore, in Figure 4 and Figure 5, “green/indigo” and “blue” were connected because the Chinese often associate them with the famous saying, “青出于蓝而胜于蓝” (Indigo is extracted from the indigo plant but is bluer than the plant from which it comes). This suggests that some lexical nodes might be associated with semantic similarities. In addition, in Figure 3, the nodes “peacock” 孔雀 [kǒnɡ què] and “dinosaur” 恐龙 [kǒng lónɡ] were associated, which might be because they shared the initial phoneme “kǒnɡ”. This suggests that lexical nodes might also become clustered according to phonological similarities. Moreover, words were also organized with morphological and syntactic similarities. For example, in Figure 2, “chicken” 鸡 [jī] and “goose” 鹅 [é] were connected because they shared the same Chinese radical “鸟” (bird).

#### 2.3.2. Network Validation

Random networks were generated with a similar number of nodes to the four lexical networks and a fixed probability (0.5) [42]. For each lexical network, 1000 random networks were generated, and their CC, ASPL, and Q index were computed. The mean values of the network parameters of 1000 random graphs were compared with the counterparts of the lexical networks. We found that the parameters of the lexical networks were significantly different from those of the random networks (CC: *t* = 95.469, *p* < 0.001; ASPL: *t* = −74.127, *p* < 0.001; Q: *t* = 44.079, *p* < 0.001). Thus, the network analysis for the study was validated.

In summary, the results of experiment one showed that the Hakka-Mandarin dialect bilingual lexicons displayed small-world properties, and the lexical information within the D2 lexicon would become transferred more efficiently. In addition, the words within the networks were organized with semantic, morphological, syntactic, and phonological similarities. However, the results could not let us understand the relations between lexicons. Thus, we conducted experiment two for exploration.

## 3. Experiment Two

To answer the second question, we used a cross-language long-term repetition priming paradigm. Experiment 2a used an animacy decision task in the study and test phases to explore the conceptual relations between D1 and D2. Experiment 2b used a lexical decision task in both phases to explore the lexical relations between the dialects. Experiment 2c used a semantic fluency task in the study phase and a lexical decision task in the test phase to explore the relations between the conceptual and lexical representations. Since Hakka natives seldom use Hakka in the orthographic form, experiment two was conducted auditorily to guarantee ecological validity.

### 3.1. Experiment 2a

Experiment 2a mainly observed whether D1 and D2 shared a conceptual representation. We assumed a shared conceptual representation if a cross-language priming effect occurred.

#### 3.1.1. Method

##### Participants

Thirty-two students (nine males) from a university participated in the study. Prior to the experiment, they completed the bilingual language profile questionnaire [37]. The results showed that participants were exposed to Hakka since birth and began learning Mandarin around the age of 4.23. Their proficiency in listening (*M*-Hakka = 5.23, *M*-Mandarin = 5.39, *t* = 0.62, *p* > 0.05) and speaking (*M*-Hakka = 5.00, *M*-Mandarin = 5.15, *t* = −0.52, *p* > 0.05) did not significantly differ between D1 and D2. They reported that they more frequently used D1 in informal settings and more frequently used D2 in formal settings. Since they were all college students who were immersed in an educational setting, they used D2 more frequently in this case.

##### Design and Materials

The experiment used a 2 (stimulus type: learned and unlearned) × 2 (language condition: D1-D2 and D2-D1) two-factor within-subject design. The D1-D2 condition refers to the situation where participants listened to Hakka words in the study phase and listened to Mandarin words in the test phase, whereas the D2-D1 condition used the reversed order. The priming effects of different language conditions and stimulus types were observed as dependent variables.

We originally selected 100 Mandarin animate words and 100 Mandarin inanimate words from previous work [2,43]. They were disyllabic words, with inanimate words selected from object, clothes, architecture, and place categories, and the animate words were selected from person, profession (such as doctor or teacher), animal, plant, fruit, and vegetable categories [2,29,43]. All these words were translated into Hakka by six Hakka native speakers from the Guangdong province, after which we invited a post-graduate student to record the materials who spoke standard Hakka and passed the Mandarin proficiency test. The audio files were 16 bits and 4800 Hz in size and were approximately 900 ms in length. Each file was edited, and its noise was reduced with Adobe Audition Software (version 6, Adobe Inc., San Jose, CA, USA). To guarantee the high familiarity of participants with the words, we invited 15 homogeneous subjects who did not join the formal experiment to rate their familiarity with the materials on a 7-point scale (1 meant not at all familiar and 7 meant very familiar).

Eventually, 40 pairs of animate words and 40 pairs of inanimate words were chosen. No significant difference was found in the familiarity between the Hakka and Mandarin words (*M*-Hakka = 6.63, *M*-Mandarin = 6.86, *t* = 1.50, *p* > 0.05). The 80 pairs of words were divided randomly into two groups, one for the D1-D2 condition (20 pairs of animate words and 20 pairs of inanimate words) and one for the D2-D1 condition. The words within each condition were further split into two groups. Each group had 10 pairs of animate words and 10 pairs of inanimate words, and they were used as the learned words and unlearned words, respectively. Moreover, 120 pairs of words were selected as fillers, with 60 pairs for the D1-D2 condition and 60 pairs for the D2-D1 condition. All the materials were counterbalanced within and between subjects, forming eight sets of materials.

##### Procedure

The experiment was carried out with E-prime 2.0 software (version 2.0, Psychology Software Tools, Pittsburgh, PA, USA) [44], and it included study and test blocks in both language conditions. Before the experiment, the participants received written instructions and practiced the trials until their accuracy rate reached 80 percent. In the formal experiment, each trial began with a fixation mark (+) occurring for 500 ms. Subsequently, a target stimulus followed, during which a question mark appeared at the center of the screen, asking participants to choose. Participants pressed the button “F” if the word they heard was animate or pressed the button “J” if the word was inanimate. The question mark would disappear if participants did not respond within 2000 ms, and the next trial would commence. All the participants went through the study and test phases in both language conditions in a counterbalanced order. The experiment took approximately 30 min. Computers automatically recorded the reaction time (RT) of the learned and unlearned words. The priming effect was determined by subtracting the RT of the learned words from that of the unlearned words [29].

#### 3.1.2. Results and Analysis

We excluded the data of the participants whose accuracy rate was lower than 80% and removed the RTs beyond ±2.5 standard deviations from the mean. Only the RT of the correct response was considered in the analysis. The means of the RTs for each language condition are summarized in Table 2, showing that the RT was faster in the learned condition than in the unlearned condition.

We employed linear mixed-effect models to analyze the data because of the strengths of a more reliable analysis of the original dataset and more generalizable findings [45,46]. The analysis was conducted with the lme4 package (version 1.1.27.1, Bates et al.) [47] of R software (version 4.1.2, R core team, Vienna, Austria) [39]. Separate analyses were conducted for each language condition (D1-D2 or D2-D1). The independent variable (stimulus type) was considered the fixed effect for each analysis. The RT was log-transformed (base e) as the dependent variable to allow the error terms to follow a normal distribution better [48]. In each analysis, we originally constructed a maximal model which included the by-subject and by-item random intercepts and random slopes for each stimulus type. Then, we used a backward stepwise selection to determine the best-fitting random effect structure.

The results are displayed in Table 2, from which we found a priming effect in the D1-D2 condition but no priming effect in the reverse condition. Though asymmetrical, the occurrence of the priming effect in the D1-D2 condition showed that D1 and D2 shared a conceptual representation.

### 3.2. Experiment 2b

Experiment 2b mainly investigated the lexical relations between D1 and D2. If no priming effect occurred, the lexical representations would be separate.

#### 3.2.1. Method

##### Participants

Forty-two students (seven males) from a university joined the experiment and finished the bilingual language profile questionnaire [37]. The results showed that the participants were exposed to Hakka since birth and began learning Mandarin around the age of 4.74. Their proficiency in listening (*M*-Hakka = 5.00, *M*-Mandarin = 5.24, *t* = 1.952, *p* > 0.05) and speaking (*M*-Hakka = 4.81, *M*-Mandarin = 4.74, *t* = 0.503, *p* > 0.05) did not differ significantly between the dialects. They reported that D1 was more frequently used in informal interactions, while D2 was more frequently used in a formal setting. Having been immersed in an educational setting, they reported using D2 more frequently in college life.

##### Design and Materials

The experimental design and stimuli were the same as that of experiment 2a. However, in experiment 2b, a lexical decision task was used in both phases, which required the participants to decide which language the words they heard belonged to. French words were chosen as fillers because French differs greatly from both Mandarin and Hakka phonologically [49]. A total of 168 disyllabic French words were constructed as fillers, with 84 French words for each language condition. All French fillers were recorded by the same speaker mentioned in experiment 2a, who had also learned French for four years. The audio files were 16 bits and 4800 Hz in size, and approximately 900 ms in length. All the materials were counterbalanced within and between subjects, forming eight sets of materials.

##### Procedure

The procedures were identical to those of experiment 2a. In both blocks, the participants were required to press the button “F” if the word they heard was French and press the button “J” if the word was Mandarin or Hakka. In the study block, 12 fillers (6 Chinese fillers and 6 French fillers) appeared first to allow the participants to familiarize themselves with the procedures, then 20 experimental stimuli and 32 fillers (6 Chinese words and 26 French words) followed. In the test block, 12 fillers appeared (6 Chinese fillers and 6 French fillers) first, followed by 40 experimental stimuli and 52 fillers (6 Chinese fillers and 46 French fillers).

#### 3.2.2. Results and Analysis

The data removal criterion was identical to that of experiment 2a. The mean RT for each condition is summarized in Table 3, which shows that the learned words were responded to more quickly than the unlearned words.

The RTs were analyzed with linear mixed-effects models. The procedures for the model construction and selection were identical to those of experiment 2a. The results are also summarized in Table 3, from which we found no significant difference in the RTs between different stimulus types. Hence no priming effect was found in both conditions, suggesting a possibility that D1 and D2 had separate lexical representations.

### 3.3. Experiment 2c

Experiment 2c mainly investigated the relations between the conceptual and lexical representations of the dialects, namely, whether the conceptual processing of one dialect could activate the lexical representation of another dialect.

#### 3.3.1. Method

##### Participants

Forty students from a university (eight males) joined the experiment and finished the bilingual language profile questionnaire [37]. They were exposed to Hakka since birth and began learning Mandarin around the age of 4.88. Their proficiency in listening (*M*-Hakka = 4.88, *M*-Mandarin = 5.10, *t* = 1.711, *p* > 0.05) and speaking (*M*-Hakka = 4.80, *M*-Mandarin = 5.00, *t* = 1.24, *p* > 0.05) did not differ significantly between the dialects. Similarly, they also reported using D2 more frequently in their college life.

##### Design and Materials

The design and materials were the same as in experiments 2a and 2b.

##### Procedure

The procedures were identical to those of experiments 2a and 2b. The participants finished the animacy decision task in the study block and the lexical decision task in the test block. The study block began with 10 fillers, followed by 20 experimental stimuli and 10 more fillers. The test block started with 12 fillers (6 Chinese fillers and 6 French fillers), followed by 40 experimental stimuli and 52 fillers (6 Chinese fillers and 46 French fillers).

#### 3.3.2. Results and Analysis

The data removal criterion was identical to that of experiments 2a and 2b. The mean RTs for each condition is summarized in Table 4.

The RTs were analyzed with linear mixed-effects models. The procedures for the model construction and selection were the same as those of experiments 2a and 2b. The results of the analysis are also summarized in Table 4, where we found no significant difference in the RTs between different types of words and, thus, no priming effect in both conditions. This indicates that the conceptual processing of one dialect may not activate the lexical representation of another dialect. This further suggests a weak link between the dialects in the lexical representation, indicating the possibility of separate lexical representations between D1 and D2.

## 4. General Discussion

The current study was the first to comprehensively investigate the mental lexicon features of Hakka-Mandarin dialect bilingual speakers from the perspectives of the structural features of lexicons and the relations between lexicons. We found that dialect bilingual lexicons were small-world in nature, and the lexical information within the D2 lexicon was transferred more efficiently than its D1 counterpart. The words within the lexicons were connected by semantic, morphological, syntactic, and phonological similarities. Additionally, we found that D1 and D2 might have a shared conceptual representation but separate lexical representations. Additionally, the conceptual processing of one dialect might not activate the lexical representation of another dialect. Based on these findings, we tentatively proposed a dialect bilingual two-layer activation model to simulate the dialect bilingual lexicon structure.

### 4.1. Structural Features of the Dialect Bilingual Lexicon

The D1 and D2 lexicons displayed small-world properties, which were consistent with the findings of previous work [10,18]. This means that, similar to bilingual lexicons, words within the dialect bilingual lexicon are closely connected, and information can travel efficiently. The small-world property also suggests the robustness of dialect bilingual lexicons [15]. For example, during novel language learning, the small-world nature of lexicons could resist the resonance from novel language to D1, and the D2 lexicons maintain the systems intact [50]. In addition, as mentioned above, we found that words within the dialect bilingual lexicon were organized by semantic, morphological, syntactic, and phonological similarities, supporting the hypothesis of the revised spreading activation model.

Moreover, the lexical information within the D2 lexicon was found to be transferred more efficiently than its D1 counterpart, which was inconsistent with the findings of previous work [10,18]. This might be because D2 is used in the educational setting. Thus, the participants tended to learn the concepts of animal and color via D2, whereas D1 was used exclusively in daily interactions, and their lexicon could not easily expand to new domains. This results in more efficient information transfer in the D2 lexicon. The finding could also be explained by language use frequency; the participants in previous studies used L1 more frequently than L2 [10,18]. Consequently, the information within the L1 lexicon was more efficiently transferred. However, the participants in the current study were college students who were immersed in an educational setting; thus, their D2 use frequency was much higher than D1. This leads to more efficient communication among the lexical nodes in D2. Nevertheless, the current study did not directly examine the role of language use frequency in the configuration of the lexicon structure, which necessitates further investigation in the future. In general, the findings indicate that, given that D2 is the official linguistic variation in China, the participants became more exposed to D2. Henceforth, D2 has increasingly influenced, assimilated, roofed over, and incorporated many linguistic aspects of D1, such as phonological and morphological aspects [51]. In other words, D1 has undergone the “corrosion” from D2, calling for a protective policy on D1. Only in this way could we restore the diversity of different variations of Chinese.

### 4.2. The Relations between Dialect Bilingual Lexicons

The finding that D1 and D2 shared a conceptual representation was in line with the results of previous work, suggesting that dialect bilingual lexicons share conceptual representations, including bilingual lexicons [2,30]. However, the priming effect was asymmetrical, with no priming effect in the D2-D1 condition. According to the RHM, the absence of such a priming effect might result from low proficiency in L2. However, the participants in the current study were equally proficient in D1 and D2, which was against the hypothesis of the RHM. The probable explanation for the asymmetrical priming effect might be that D1 and D2 had a partially shared conceptual representation, aligning with the concept of the sense model. To be specific, the absence of a priming effect resulted from fewer conceptual senses in D2 than in D1 because D2 was less dominant than D1. Nevertheless, since language dominance incorporates language proficiency and language use frequency [24], D2 in the current study should be more dominant because the participants were equally proficient in two dialects and used D2 more frequently. The inconsistency between the hypothesis of the sense model and the current case might be the different populations in focus. The sense model mainly focused on bilinguals who tend to frequently use their more proficient language in daily life, leading to language dominance. Whereas the dialect bilingual speakers examined in the current study were immersed in an educational setting which necessitates more frequent use in D2. This limits their freedom to use their more proficient mother tongue, D1, counterbalancing the dominance between the dialects. Consequently, the modulatory power of language dominance in the conceptual representation was undermined in the dialect bilingual case, which was also pointed out in previous work [2]. Instead, we assumed that the age of acquisition might be a more influential factor in the conceptual representation of the dialect bilingual lexicon. This is because the age of acquisition was found to be a determining factor in the configuration of the conceptual representation by many bilingual studies [31]. Given that the participants in the current study acquired D1 earlier than D2, the conceptual senses of D1 were more entrenched than D2, leading to more senses in D1 than in D2. This resulted in the asymmetrical priming effect. However, it is just an assumption. Future studies could empirically examine the role of the age of acquisition in the conceptual representation of the dialect bilingual lexicon.

The lexical representations of D1 and D2 were found to be possibly separate, conforming to the assumption of the RHM and sense model. However, it seems implausible because D1 and D2 both originated from ancient Chinese, and both use the Han writing system to symbolize sounds. In fact, Hakka and Mandarin share many cognates [1]. Thus, in this case, the concept of lemma representation of the WEAVER++ model would be more appropriate. Specifically, the shared lemmas between D1 and D2 within the lemma representation represent cognates. For example, the lexical forms “水角” and “水杯” share the lemma “水” (water) to describe the concept of “cup” in Hakka and Mandarin, respectively. At the same time, the other lemmas within the representation signify the unique lemmas belonging either to D1 or D2. For example, the lexical form “黦” refers to a rotted plant in Hakka [35], but it is scarcely used in Mandarin to refer to any common concepts. All the lemmas within the lemma representation would become activated to produce different lexical forms, as evidenced by the findings of experiments 2b and 2c, even in the case of cognates. For example, although the aforementioned “水角” and “水杯” both share the lemma “水”, the lexical forms are divergent in the two dialects. However, the current study did not directly examine the cognate effect in the lexical representation of the dialect bilingual lexicon, which merits further exploration in future studies.

Taken together, although the concept of the sense model seems to be plausible in explaining the conceptual relations between the dialect bilingual lexicons, it was unable to explain the absence of the priming effect in the D2-D1 direction. In addition, the sense model did not consider the cognates shared by D1 and D2, undermining its explanatory power in lexical relations between lexicons. In this case, WEAVER++ seems to be more appropriate, but it did not provide a comprehensive discussion on the conceptual representation and the interaction between the dialects in the conceptual and lexical representations. However, the dialects spoken by dialect bilingual speakers might share more similarities than bilinguals in terms of many aspects, especially the semantic and syntactic features, since they both belong to the same language and originate from the same culture [52]. The lack of focus on the interaction between the lexical and conceptual representations might, to a degree, undermine the explanatory power of the existing bilingual models. Since the semantic and syntactic properties of words are essential in sentence merging, the minute difference in terms of these lexical aspects might cause a huge difference in the interactions between lexicons and other levels of language representation in the human mind. Thus, considering the feature of dialect bilinguals might increase the plausibility of the current models. Furthermore, the bilingual models did not discuss the lexicon structure from the perspective of network science, which may hinder the models from approximating cognitive processing [13]. All these potential limitations might call for the further development of the current bilingual model to better our understanding of the bilingual mental lexicon. Therefore, based on the previous bilingual models and the results of the current study, we tentatively proposed the dialect bilingual two-layer activation model to simulate the lexicon features of dialect bilinguals.

### 4.3. Dialect Bilingual Mental Lexicon Structure

Language systems, especially mental lexicons, could be seen as a multiplex network [53]. Based on the assumptions and the findings of the current study, we assume that dialect bilingual lexicons may be a multiplex network. To simulate the dialect bilingual mental lexicon structure, we tentatively put forward a dialect bilingual two-layer activation model whose details are illustrated in Figure 6.

#### 4.3.1. Model Structure

As displayed in Figure 6, the model assumes that the D1 and D2 lexicons each incorporate a lexical and conceptual representation. The lexical representation includes the lexical form and lemma representations. Within the lexical form representation, the D1 and D2 representations are separate, with the black space referring to the D1 space and the white space referring to the D2 space. The nodes within the lexical form representation refer to the lexical nodes, and edges refer to the phonological and morphological similarities. Within the lemma representation, the black space refers to D1, the white space refers to D2, and the gray space refers to the overlapped space between D1 and D2. In other words, D1 and D2 are partially shared in this representation. Since lemmas incorporate the syntactic information of words, the nodes within the representation represent the lemma, and the edges represent the syntactic similarities. The lemma-overlapped space encapsulates the cognates shared by D1 and D2. Moreover, in the lexical representation, language use frequency may play a significant role. To be specific, a more frequently used dialect would obtain a better structure with more space, denser neighborhoods, and more modular communities. Consider the results of the current study; for example, since the participants used D2 more frequently, their D2 lexicon displayed better organization with more nodes, a tighter word association, and more lexical clusters. With language use frequency as a modulator, the lexical representation network would change dynamically over time [54].

Within the conceptual representation, the black space refers to the D1 representation, the white space refers to the D2 representation, and the gray space refers to the overlapped space between D1 and D2. The nodes refer to the conceptual nodes, and the edges refer to the semantic similarities. Unlike the lexical representation, language use frequency in this layer does not play a role. Rather, the age of acquisition may act as a more influential factor in shaping the configuration. Since the participants in the current study acquired D1 earlier than D2, their D1 conceptual representation occupied more senses than its D2 counterpart, leading to a larger semantic space in D1. The conceptual representation is assumed to be relatively stable due to the earlier acquisition of concepts in D1 in contrast to D2. Furthermore, though no connection between one dialect conceptual representation and another dialect lexical representation was found in experiment 2c, the lexical representation of the dialects would relate to their corresponding conceptual representation, allowing for mutual information transfer [53]. This is signified by bidirectional dash lines between the representations in Figure 6.

#### 4.3.2. Working Mechanism

Based on the concept of WEAVER++, the model also tried to simulate the mechanism underlying lexical perception and lexical production in the dialect bilingual case. Specifically, during lexical perception, the input activates the relevant nodes in the lexical form representation. The activation would then pass onto the corresponding node within the lemma representation, after which the semantic node in the conceptual representation would become activated to retrieve meaning. Given that, in the current case, more senses were in the D1 conceptual representation than in its D2 counterpart, the semantic information activated in D1 could travel to its overlapped space with D2, leading to a priming effect in the D1-D2 condition. However, the senses in D2 were less than in D1, rendering activation hard to travel to the D1 space, resulting in the absence of a priming effect in the D2-D1 condition.

During lexical production, the semantic node in the conceptual representation would first be targeted, which would further activate the relevant node in the lemma representation. Subsequently, the activation would become transferred to the lexical form representation [53]. Then, the combined efforts of the conceptual, lemma, and lexical representations would produce a word form that incorporates semantic, syntactic, morphological, and phonological information. This serves as the reason why in the current case, words within lexicons became connected with semantic, syntactic, morphological, and phonological similarities.

Furthermore, the model explains why we found a more efficient organization in the D2 lexicon with language use frequency as a modulator in shaping the lexicon structural features, while we found more semantic senses in the D1 conceptual representation with the age of acquisition as a modulator in shaping the conceptual configuration. These seemingly contradictory results could be accounted for by the hypothesis that the dialect bilingual lexicon exists as a multi-layer structure with different layers storing different kinds of information. Thus, naturally, different factors play distinct roles in different layers. Given that a conceptual representation only stores semantic information while the lexical representation incorporates the lexical form lemma representations, which include phonological, morphological, and syntactic information, the multi-fold information stored in the lexical representation becomes more representative of the whole lexicon. Consequently, a more efficient organization in the D2 lexicon would be natural, as evidenced by the current study.

## 5. Conclusions

Fruitful findings were obtained on the bilingual mental lexicon, while less is understood about the dialect bilingual mental lexicon. As a variation of bilinguals, dialect bilingual speakers are characteristically different from bilinguals, but their different features have not been fully considered in the model construction of the bilingual lexicon. To help understand the dialect bilingual lexicon, we investigated the topic from the perspectives of the structural features of lexicons and the relations between lexicons. We found that the dialect bilingual lexicons displayed small-world properties and the lexical information within D2 was more efficiently transferred. Additionally, D1 and D2 were found to be partially shared in the conceptual representation, separate in the lexical representation, and partially shared in the lemma representation. Based on the findings, we put forward a dialect bilingual two-layer activation model to simulate the dialect bilingual lexicon structure. The model adopts the notion of network science and specifies the role of language use frequency in a lexical representation and the role of age of acquisition in a conceptual representation, which enables the model to explain the results of the current study and some findings from previous work.

However, the paper has the following limitations. First, the language use frequency was not properly measured, and the effect of the age of acquisition on the lexicon structure was not examined directly in the current study. The roles of the two factors in different representations should be examined further in the future. Second, we mentioned the role of cognates in lexical representations, but we did not directly examine the cognate effect in the dialect bilingual lexicon. More exploration is needed to directly investigate the cognate effect on the lexicon organization of dialect bilingual speakers. Third, the experimental stimuli in experiment 2 were not well controlled for phonetic, morphological, and syntactic features. More studies are needed to understand the role of these lexical aspects in dialect bilingual lexicon features. Fourth, the findings were mainly obtained with the experiments in the auditory channel. We have no idea whether similar result patterns would be obtained with the experiment in the visual channel. Thus, research with a visual channel design could be conducted to reveal more concerning this issue. Fifth, although we obtained the priming effect in experiment 2a, the effect size was small. More studies could be conducted to replicate the findings. Moreover, the model was merely drawn from the evidence from Hakka-Mandarin speakers; more studies are needed to examine whether it could be generalized to other types of dialect bilingual speakers. Taken together, a significant amount of work is needed to be conducted before we can grasp a basic understanding of the dialect bilingual lexicon. Anyhow, this study brings some insights into the understanding of dialect bilingual lexicon features and leads us to reconsider the generalizability and applicability of existing bilingual models.

## Figures and Tables

**Figure 1 brainsci-12-01629-f001:**
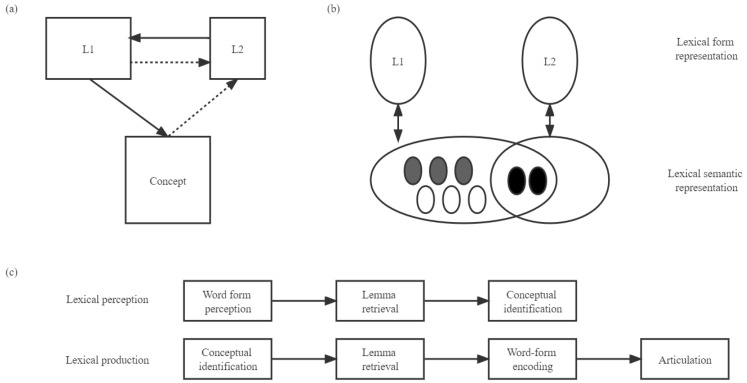
The details of the bilingual models. Note. (**a**) The revised hierarchical model. From the “Category interference in translation and picture naming: Evidence for asymmetric connection between bilingual memory representations” by J. Kroll and E. Stewart, 1994, *Journal of Memory and Language, 33* (2), 149–174. L1 refers to the L1 lexical representation, L2 refers to the L2 lexical representation, and the concept refers to the conceptual representation shared by the L1 and L2 lexical representations. (**b**) The sense model. From “The role of polysemy in masked semantic and translation priming” by M. Finkbeiner et al., 2004, *Journal of Memory and Language*, *51* (1), 1–22. L1 and L2 refer to the lexical representation of L1 and L2. The white and light grey circles within the semantic representation refer to the specific senses belonging to L1, and the dark grey circles refer to the shared senses between L1 and L2. (**c**) The WEAVER++ model. Adapted from “Goal-referenced selection of verbal action: Modeling attentional control in the Stroop task” by A. Roelofs, 2003, *Psychological Review*, *110* (1), 88–125.

**Figure 2 brainsci-12-01629-f002:**
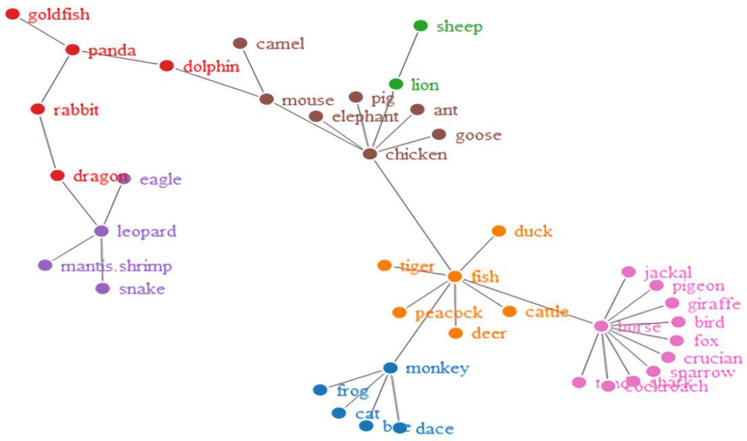
Network visualization of the Hakka-animal lexical network.

**Figure 3 brainsci-12-01629-f003:**
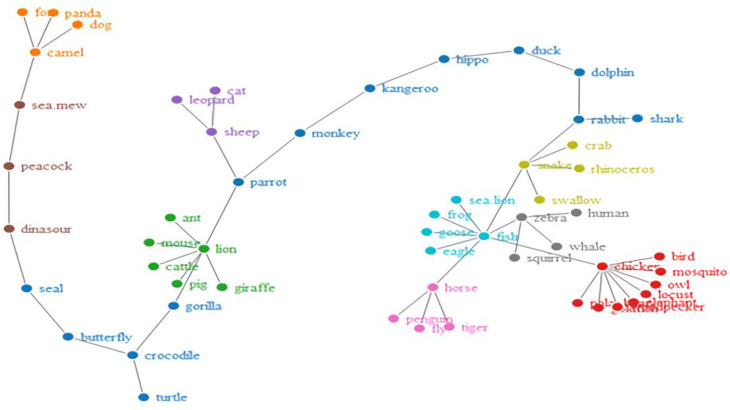
Network visualization of the Mandarin-animal lexical network.

**Figure 4 brainsci-12-01629-f004:**
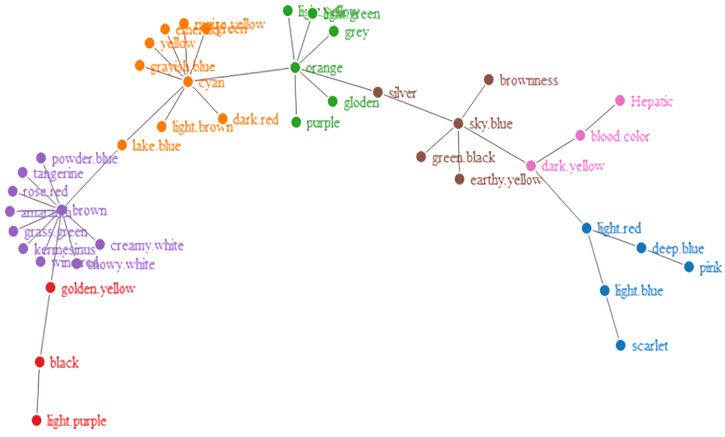
Network visualization of the Hakka-color lexical network.

**Figure 5 brainsci-12-01629-f005:**
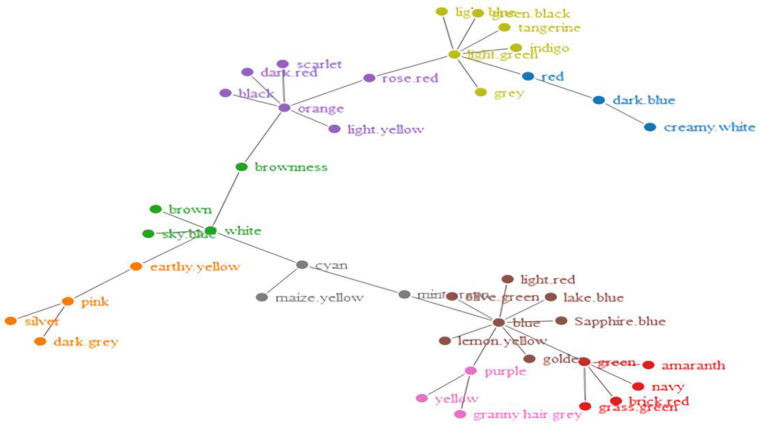
Network visualization of the Mandarin-color lexical network.

**Figure 6 brainsci-12-01629-f006:**
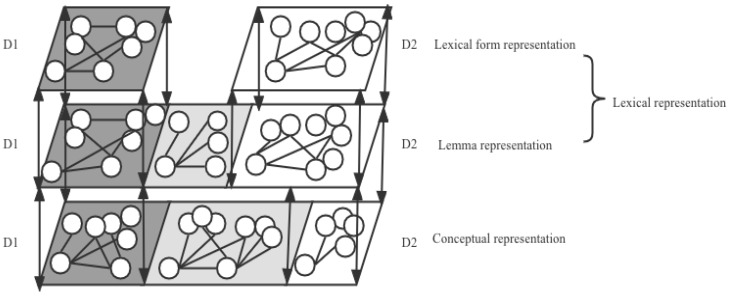
Dialect bilingual two-layer activation model.

**Table 1 brainsci-12-01629-t001:** Lexical network parameters.

N		ASPL	ASPLran	CC	CCran	S	Q
Hakka-animal	40	1.024	1.355	0.979	0.951	1.362	0.731
Mandarin-animal	55	1.016	1.363	0.986	0.966	1.370	0.782
Hakka-color	41	1.006	1.352	0.994	0.970	1.377	0.720
Mandarin-color	41	1.037	1.359	0.970	0.940	1.353	0.722

Note. N = number of nodes; ASPL = average shortest path length; ASPLran = average shortest path length of a random network; CC = clustering coefficient; CCran = clustering coefficient of a random network; S = small-world index; Q = modularity index.

**Table 2 brainsci-12-01629-t002:** Reaction times and the mixed-effects modeling results for experiment 2a.

Language Condition	Mean (SD) of the Unlearned Stimulus	Mean (SD) of the Learned Stimulus		Priming Effect	
D1-D2	395.113 (289)	389.677 (340)		5.435	
D2-D1	444.996 (345)	427.850 (345)		17.146	
Fixed effect					
Language condition		Estimate	*SE*	*t*	*p*
D1-D2	(intercept)	5.517	0.099	55.991	<0.001 ***
	Stimulus type	0.120	0.048	2.477	0.013 *
D2-D1	(intercept)	5.675	0.104	54.441	<0.001 ***
	Stimulus type	0.066	0.043	1.539	0.124

* *p* < 0.05, *** *p* < 0.001.

**Table 3 brainsci-12-01629-t003:** Reaction times and the mixed-effects modeling results for experiment 2b.

Language Condition	Mean (SD) of the Unlearned Stimulus	Mean (SD) of the Learned Stimulus		Priming Effect	
D1-D2	372.725 (272)	369.599 (230)		3.126	
D2-D1	342.912 (221)	347.909 (247)		−4.996	
Fixed effect					
Language condition		Estimate	*SE*	*t*	*p*
D1-D2	(intercept)	5.719	0.076	75.150	<0.001 ***
	Stimulus type	−0.006	0.039	−0.162	0.871
D2-D1	(intercept)	5.621	0.083	67.403	<0.001 ***
	Stimulus type	0.006	0.046	0.126	0.900

*** *p* < 0.001.

**Table 4 brainsci-12-01629-t004:** Reaction times and the mixed-effect modeling results for the experiment.

Language Condition	Mean (SD) of the Unlearned Stimulus	Mean (SD) of the Learned Stimulus		Priming Effect	
D1-D2	372.356 (267)	375.225 (255)		−2.868	
D2-D1	397.428 (260)	411.796 (293)		−14.368	
Fixed effect					
Language condition		Estimate	*SE*	*t*	*p*
D1-D2	(intercept)	5.724	0.090	63.639	<0.001 ***
	Stimulus type	−0.024	0.038	−0.632	0.528
D2-D1	(intercept)	5.845	0.083	70.344	<0.001 ***
	Stimulus type	−0.034	0.035	−0.973	0.331

*** *p* < 0.001.

## Data Availability

Data are available from the corresponding author with a request.

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
