# Peer review of "The Mental Lexicon Features of the Hakka-Mandarin Dialect Bilingual"

_brainsci, 2022, doi:10.3390/brainsci12121629_

Round 1

Reviewer 1 Report

The paper is well-written both in style and organization. I think it fulfills the standards of papers of that kind. It supports what a sociolinguist might expect: a relative cognitive weakening of the uncodified variety in today's "bilingual speakers".

–– Very rarely, sentences are more difficult to understand; I point out one: PAGE 002 LINE 075-078: "Complex ..." > ", complex ..." (?)

I am not myself a psycholinguist in the sense of this paper; therefore, I will restrict my commentaries to the sociolinguistic aspects:

–– "bidialectal": While I do accept your viewpoint to be presented as such, and it won't change anything about the study, one might attempt to define the words "bilingual" vs. "bidialectal" precisely; the Chinese view on "fangyan" is not exactly the same as the European view on "dialect" insofar as the differences between Hakka and Mandarin Chinese could be seen as big as between Dutch and German. In other words, I am wondering whether it is not "bilingualism of fairly closely related languages". [The comparison is not precise, because both German and Dutch are standardized languages; maybe I should say "Flamish (Southern Dutch) vs. German": There is considerable overlap, phonological similarities, but then also unbridgeable differences; nonetheless, speakers might quickly adapt to each other if necessary.] – I certainly agree that Hakka is (historically) closer to Mandarin than, e.g., Cantonese or Hokkien; however, "Mandarin" speakers usually won't understand a word.

–– This leads to another point: You are presenting the difference as "speaking two dialects": Hakka is a local spoken variety, while what is called Mandarin here is most probably a competence in Putonghua, i.e., in the STANDARD language taught in school – and not some "Mandarin dialect". There is an asymmetry between spoken, uncodified languages and standardized, codified languages – especially in the lexicon: While uncodified languages are learned and used exclusively in daily interactions and cannot easily expand their lexicon to new domains, the standard language is planned, extensible, supported by lexical enrichment and a written form, and taught at schools. While dialects would have dominated in a region earlier, in the modern situation, with good general education and full access even to spoken forms of a "national (codified) language", they are (generally) quickly becoming (cognitively) dominant over all other idioms. That supports the outcome of your study. However, what you are comparing is a dominant language vs. a local language (dialect), not precisely "two dialects" (as if they were of the same sociolinguistic value).

The influence of a standard upon "closer" spoken varieties ("dialects") and more remote speech varieties ("minority languages") has been discussed in Kloss ("linguistic distance"; see below). In short, Kloss (and others) observed how dominating languages begin to increasingly influence vernaculars, "roof over" them, and then "incorporate" them as "dialects" or "sociolects".

–– Hakka can be written with Chinese characters, for sure, however, this is most probably a non-codified use of writing (except for academics working on Hakka, or traditionally, when a spoken standard was not available, i.e., writing and speaking were not strictly associated). One needs to evaluate whether there is a certain bias from using the characters with people otherwise perhaps not used to reading in Hakka, but being well-versed in Putonghua.

These are just general thoughts which cannot be terminally discussed in this paper, but which I would contemplate.

One can also take your view and say that it is interesting to see how different "dialects" of a "language" may sometimes be. After all, the definition of "a language" is virtually impossible.

In any case, I am interested to see such research which supports sociolinguistic observations from a more experimental approach.

KLOSS - REFERENCES

Kloss, H. 1967. Abstand-languages and Ausbau-languages. An­thro­pological Lin­guistics 9: 29-41. ["distance languages" and "elaboration-languages"]

(unfortunately in German:)

Kloss, H. 1978 [1952]. Die Entwicklung neuer germanischer Kul­tur­sprachen seit 1800. 2. erw. Aufl., Düsseldorf.
Kloss, H. 1987. Abstandsprache und Ausbausprache. In U. Am­mon and N. Dittmar and K.J. Mattheier (eds.), Sozio­lin­guis­tik. Handbuch zur Wissenschaft von Sprache und Ge­sell­schaft 1, pp. 302-315. Berlin.

Author Response

Dear reviewer,

Thank you much for your insightful comments, they are very helpful in our revision. And we responded to your comments as follows.

1. Very rarely, sentences are more difficult to understand; I point out one: PAGE 002 LINE 075-078: "Complex ..." > ", complex ..."

Sorry for this. We modified the language and made it more clear in line 76-line 80 on page 2.

2. "bidialectal": While I do accept your viewpoint to be presented as such, and it won't change anything about the study, one might attempt to define the words "bilingual" vs. "bidialectal" precisely; the Chinese view on "fangyan" is not exactly the same as the European view on "dialect" insofar as the differences between Hakka and Mandarin Chinese could be seen as big as between Dutch and German. In other words, I am wondering whether it is not "bilingualism of fairly closely related languages". [The comparison is not precise, because both German and Dutch are standardized languages; maybe I should say "Flamish (Southern Dutch) vs. German": There is considerable overlap, phonological similarities, but then also unbridgeable differences; nonetheless, speakers might quickly adapt to each other if necessary.] – I certainly agree that Hakka is (historically) closer to Mandarin than, e.g., Cantonese or Hokkien; however, "Mandarin" speakers usually won't understand a word.

Thanks for pointing this out. However, as for Dutch and German, no matter how similar they are in terms of phonological or morphological patterns, each of the language has it own standardized writing system. And the writing systems differed between Dutch and German. While it is not the case for Hakka. Hakka has no codified writing system, and it can only be symbolized with Han writing system (the same writing system for Mandarin). Besides, according to the Language Atlas of China (Institute of Linguistics, 2012), Hakka and Mandarin are defined as different variations of Chinese together with Min, Cantonese and other variations. These variations are referred to different dialects within Chinese. Thus, Hakka cannot be defined as a language. We still regard Hakka and Mandarin as dialects.

Reference:

Institute of Linguistics. (2012). Language Atlas of China - Chinese Dialects. The Commercial Press.

3. This leads to another point: You are presenting the difference as "speaking two dialects": Hakka is a local spoken variety, while what is called Mandarin here is most probably a competence in Putonghua, i.e., in the STANDARD language taught in school – and not some "Mandarin dialect". There is an asymmetry between spoken, uncodified languages and standardized, codified languages – especially in the lexicon: While uncodified languages are learned and used exclusively in daily interactions and cannot easily expand their lexicon to new domains, the standard language is planned, extensible, supported by lexical enrichment and a written form, and taught at schools. While dialects would have dominated in a region earlier, in the modern situation, with good general education and full access even to spoken forms of a "national (codified) language", they are (generally) quickly becoming (cognitively) dominant over all other idioms. That supports the outcome of your study. However, what you are comparing is a dominant language vs. a local language (dialect), not precisely "two dialects" (as if they were of the same sociolinguistic value).

Thanks for the suggestions. It is helpful in explaining our results, and we modified the discussion of the findings with the suggestions into consideration (details can be found in line 552-line 557 on page 14). Nevertheless, according to the Language Atlas of China (Institute of Linguistics, 2012), Hakka and Mandarin (Guanhua) are defined as different variations of Chinese together with Min, Cantonese and other variations. Therefore, we still consider Hakka and Mandarin as two dialects in the current paper. Besides, in psycholinguistics, we define bilinguals as the speakers of two different languages, but bidialectal speakers as the speakers of two dialects. Thus, here in the case of the current study, since Hakka and Mandarin were not two different languages, we preferred to refer to Hakka-Mandarin speakers as bidialectal speakers rather than bilinguals.

Reference:

Institute of Linguistics. (2012). Language Atlas of China - Chinese Dialects. The Commercial Press.

4. The influence of a standard upon "closer" spoken varieties ("dialects") and more remote speech varieties ("minority languages") has been discussed in Kloss ("linguistic distance"; see below). In short, Kloss (and others) observed how dominating languages begin to increasingly influence vernaculars, "roof over" them, and then "incorporate" them as "dialects" or "sociolects".

Thank you for recommending this. We went through them and modified our discussion of the relevant results according to the opinions of the papers (details can be found in line 562-line 568 on page 14-15).

5. Hakka can be written with Chinese characters, for sure, however, this is most probably a non-codified use of writing (except for academics working on Hakka, or traditionally, when a spoken standard was not available, i.e., writing and speaking were not strictly associated). One needs to evaluate whether there is a certain bias from using the characters with people otherwise perhaps not used to reading in Hakka, but being well-versed in Putonghua.

Yes, thank you for pointing out this. However, the experiment was conducted in auditory channel. Given that Hakka is seldom used in reading, we designed the auditory experiment to explore the lexicon structure of the bidialectal speakers. We thought that, in this way, we could relatively guarantee the ecological validity of the experiment. Therefore, whether the participants are used to reading in Hakka might not bring much influence to the findings since it was in auditory channel. However, what you pointed out is very interesting, and it would be true if the experiments were conducted in visual channel. Thus, we listed it as the direction for future research in the conclusion section.

Reviewer 2 Report

This study investigated the mental lexicon of Hakka-Mandarin bidialectal speakers. The experimental results suggested that the D2 lexicon of Mandarin is better organized, possibly due to it being the dominant language. In addition, the two dialects are likely to share parts of the conceptual representations, but have separate lexical representations. As these results cannot be fully accounted for by the existing models, the authors proposed a novel bidialectal lexicon model.

I found the issue of bidialectal lexicon very interesting, as well as theoretically important. I liked the way that the authors addressed this issue with experimental paradigms and with careful data analyses. The manuscript is very well structured and clearly written, and is easy to follow. However, I have some conceptual and methodological concerns about this paper:

1. Although I understand that the distinction between language and dialect is difficult, and involves various considerations, I however do not agree with the authors that Hakka and Mandarin should be treated as two dialects simply becasue they both originated from ancient Chinese and that they have a lot of cognates. Cognates are found in many languages as well, and many languages are closely related to each other in terms of their historical origins. Furthermore, treating two mutually unintelligible languages as dialects does not do justice to the dialects that are mutually intelligible, such as British English and American English. I therefore think that it is better to adopt commonly accepted linguistic standard for language/dialect distinction, by which Hakka and Mandarin should be two different languages, not dialects.

2. Following my previous point, terminology aside, I still found it theoretically interesting to study Hakka-Mandarin bilingual speakers. As the authors mentioned, these two languages share a lot of cognates. I therefore had expected some analyses or comparison between cognates and non-cognates, as this has been an important topic in the bilingual literature (e.g., Dijkstra, T., Miwa, K., Brummelhuis, B., Sappelli, M., & Baayen, H. (2010). How cross-language similarity and task demands affect cognate recognition. Journal of Memory and language, 62(3), 284-301.) I think this paper will be theoretically stronger if the cognate effect can be discussed.

3. I truly appreciate the authors' rigid experimental design and detailed statistical analyses. Nevertheless, I have some concerns with regards to the experimental results. For all the long-term priming experiments (Experiment 2), there was only one signficant effect (Experiment 2a, D1-D2), and the effect size was very small. As the authors' conclusion is based on a lot of null results, I would stronly recommend the authors to run further replication experiments to see whether the results can be replicated.

4. Specifically for Experiment 1, probably I have missed something, but I did not understand how the authors arrived at the conclusion that "words within networks got organized with semantic, morphological, syntactic and phonological similarities". Especially for Figures 2-5, there is no explanation for the color coding, and the resolution of the figures is very bad, so that the words in the figures cannot be seen clearly. More clarification will be needed for these figures.

Author Response

Dear reviewer,

Thank you very much for your insightful comments. They are really helpful in the revision of our manuscript. We responded to your comments as follows.

1. Although I understand that the distinction between language and dialect is difficult, and involves various considerations, I however do not agree with the authors that Hakka and Mandarin should be treated as two dialects simply because they both originated from ancient Chinese and that they have a lot of cognates. Cognates are found in many languages as well, and many languages are closely related to each other in terms of their historical origins. Furthermore, treating two mutually unintelligible languages as dialects does not do justice to the dialects that are mutually intelligible, such as British English and American English. I therefore think that it is better to adopt commonly accepted linguistic standard for language/dialect distinction, by which Hakka and Mandarin should be two different languages, not dialects.

Thank you for pointing this out. However, we still regard Hakka and Mandarin as two dialects. This is not only because they originate from ancient Chinese and share lots of similarities, but also because they are different variations of Chinese. According to the Language Atlas of China (Institute of Linguistics, 2012), Hakka and Mandarin are defined as different variations of Chinese together with Min, Cantonese and other variations. These variations are referred to as different dialects within Chinese context. Besides, Hakka has no codified writing system, and it can only be symbolized with Han writing system. This distinguishes Hakka as a dialect, not a different language with totally different writing system. Moreover, although Hakka and Mandarin are not mutually intelligible, we can not define them as different languages simply with the standard of mutual intelligibility. This is because some dialects are mutually intelligible owing to relatively close geographical distance and historical factors while some dialects are not mutually intelligible due to the accumulative farther geographical distance. Thus, the mutual intelligibility should not be the only standard for the definition of dialects. As for the different languages in China, there are Uygur language, Tibetan and Mongolian. That these languages can be defined as different languages rather than dialects is because they have codified writing system and they have different phonological and morphological systems from Mandarin (Han). Taken together, we still think that we are exploring the mental lexicon of bidialectal speakers rather than bilinguals.

Reference:

Institute of Linguistics. (2012). Language Atlas of China - Chinese Dialects. The Commercial Press.

2. Following my previous point, terminology aside, I still found it theoretically interesting to study Hakka-Mandarin bilingual speakers. As the authors mentioned, these two languages share a lot of cognates. I therefore had expected some analyses or comparison between cognates and non-cognates, as this has been an important topic in the bilingual literature (e.g., Dijkstra, T., Miwa, K., Brummelhuis, B., Sappelli, M., & Baayen, H. (2010). How cross-language similarity and task demands affect cognate recognition. Journal of Memory and language, 62(3), 284-301.) I think this paper will be theoretically stronger if the cognate effect can be discussed.

Thank you for the brilliant suggestion. However, unfortunately, we did not examine the effect of cognate in the research design of the current study. Thus, we could only mention a little bit in the discussion section (line 600-line 616 on page 15). We listed it as the limitation of the current study and pointed it out for future research.

3. I truly appreciate the authors' rigid experimental design and detailed statistical analyses. Nevertheless, I have some concerns with regards to the experimental results. For all the long-term priming experiments (Experiment 2), there was only one significant effect (Experiment 2a, D1-D2), and the effect size was very small.As the authors' conclusion is based on a lot of null results, I would strongly recommend the authors to run further replication experiments to see whether the results can be replicated.

Thank you for pointing this out. We only obtained the priming effect in Experiment 2a, suggesting that two dialects might share conceptual representation. This aligned with the previous findings. However, we did not obtain priming effect in Experiment 2b and 2c, suggesting that the lexical representations might be separate in a degree. Based on all the findings, we believe that the absence of priming effect found in Experiment 2b and 2c is still meaningful. Sometimes, null effects can also be meaningful in the interpretation of the findings. But your point of view is reasonable concerning the small effect size obtained in Experiment 2a, thus we listed it as a weakness of the current study in the conclusion section and called for more research to replicate the findings.

4. Specifically for Experiment 1, probably I have missed something, but I did not understand how the authors arrived at the conclusion that "words within networks got organized with semantic, morphological, syntactic and phonological similarities". Especially for Figures 2-5, there is no explanation for the color coding, and the resolution of the figures is very bad, so that the words in the figures cannot be seen clearly. More clarification will be needed for these figures.

Thanks for the suggestions. We added more clarification of the figures in the paper, and improved the clarity of these figures. The details can be found in line 321-line 345 on page 7-9.

Reviewer 3 Report

The manuscript proposes a new model that simulates mental lexicon of bidialectal speakers based on data obtained from a series of behavioral experiments that examined mental lexicon features of Hakka-Mandarin bidialectal speakers. The project seeks to fill a known gap in the bilingual studies: i.e., examination of interactions of dialects within one individual that share more properties then two different languages in a typical bilingual speaker. I could recommend the manuscript for publication with Brain Sciences journal pending the authors address the following points:

1.     P7. Line 300 – every time the authors talk about D2 versus D1 lexicon organization comparisons please refrain from using the term “X is better organized then Y” – “better”  is a very vague term. What makes it “better? Instead, please use more descriptive adjectives that reflect the decision-making process that authors used to compare the lexicons: i.e. unpack the meaning of ASPL, CC, S and Q clusters – e.g. “Lexicon D2 entries have shorter ASPL values then Lexicon D1 – give the numbers from the table and that means…give the interpretation here”.  The readers need help interpreting parameter values form Table 1 as the values for Hakka and Mandarin look very close to each other. 

2.     I have very serious concerns regarding the lme analyses reported in Table 2:

a.     Please double check that the coefficients and t values are not flipped between the two models. If the values are indeed as they are reported the authors need to explain why 5 millisecond difference between D1-D2 was significant but 17 millisecond difference between D2-D1 was not. It does not make any sense. What would help here is plotting individual means to show what the actual human results looked like. 

3.     Final point concerns limitations – please add that current stimuli in Experiment 2 where not controlled for phonetic, morphologic, and syntactic features and as such more studies are needed to understand the role of these lexical aspects in bidialectal lexicons. 

Author Response

Reply to Reviewer Report

  1. Line 300 – every time the authors talk about D2 versus D1 lexicon organization comparisons please refrain from using the term “X is better organized then Y” – “better” is a very vague term. What makes it “better? Instead, please use more descriptive adjectives that reflect the decision-making process that authors used to compare the lexicons: i.e. unpack the meaning of ASPL, CC, S and Q clusters – e.g. “Lexicon D2 entries have shorter ASPL values then Lexicon D1 – give the numbers from the table and that means…give the interpretation here”. The readers need help interpreting parameter values form Table 1 as the values for Hakka and Mandarin look very close to each other.

Thanks for the valuable suggestions, according to which we modified the part mentioned by the reviewer (details can be found in the second paragraph of section 2.3.1) and the rest of the paper. Specifically, we used the expression “the lexical information within D2 lexicon got more efficiently transferred” rather than “D2 lexicon was better-organized”.

  1. I have very serious concerns regarding the lme analyses reported in Table 2:
    1. Please double check that the coefficients and t values are not flipped between the two models. If the values are indeed as they are reported the authors need to explain why 5 millisecond difference between D1-D2 was significant but 17 millisecond difference between D2-D1 was not. It does not make any sense. What would help here is plotting individual means to show what the actual human results looked like.

Thanks for pointing this out. We carefully double-checked the data and redid the analysis, and the results remained the same. Besides, we found that, according to the results of the best-fitting model, the estimate in the D1-D2 condition was larger than the one of the reversed condition (as shown in Table 2, Estimate_D1-D2=0.120, Estimate_D2-D1=0.066). This suggests that the slope of changes in the D1-D2 condition is larger than the counterpart in the D2-D1 condition, indicating that the reaction time difference in D1-D2 condition is more likely to get significant. This might also get evidenced from the individual means graphs (the plots are displayed in the appendix which is attached to the reply to the review report).

Concerning why we obtained significance in D1-D2 even if the priming effect is small and why we did not obtain significance in the reversed condition even if the priming effect is large, we think this might be because of standard deviation. As we can see in Table 2, in the D1-D2 condition, the standard deviation of the unlearned stimulus was 289, and the one of the learned stimuli was 340. Whereas, in the D2-D1 condition, the standard deviation of the unlearned stimulus was 345, and the one of the learned stimuli was also 345. The difference in standard deviation between different types of stimulus in the D1-D2 condition was larger than the counterpart in the D2-D1 condition. This might have probably influenced the results we obtained. To facilitate the evaluation of this part, we attached the data and relevant codes in the appendix attached in the reply to the review report.

  1. Final point concerns limitations – please add that current stimuli in Experiment 2 where not controlled for phonetic, morphologic, and syntactic features and as such more studies are needed to understand the role of these lexical aspects in bidialectal lexicons.

Thanks for the suggestion. We added the relevant limitation in the conclusion section (details can be found in line 697-line 699).

Round 2

Reviewer 2 Report

I appreciate the authors' quick response and revision. For the language-dialect distinction, I agree that this is an issue that should be open to discussion, and I fully respect the authors' view on this issue. However, I am still not convinced that Mandarin and Hakka should be treated as two dialects. It is true that mutual intelligibility should not be the only criterion for defining dialects, but it is also not clear to me why in the case of Hakka and Mandarin "political and social convention" is a better criterion, especially when the focus of this paper is on the cognitive perspective of mental lexicon, not on the social and culture perspectives. In addition, If the authors try to propose a model for bidialectal mental lexicon, which is supposed to be different from a bilingual model, the authors will need to explain more clearly why the current results can only be accounted for by assuming that the speakers speak two dialects, instead of two languages that are rather similar and share many cognates.

With respect to the experimental results, I still consider the small effect size to be very worrisome. Given that the model that the authors proposed is completely based on these results, from my perspective, replication experiments are indispensable. However, this is my personal opinion, which is very likely to be biased. I will therefore leave the decision to the editor.

Author Response

The reply to the review report

  1. However, I am still not convinced that Mandarin and Hakka should be treated as two dialects. It is true that mutual intelligibility should not be the only criterion for defining dialects, but it is also not clear to me why in the case of Hakka and Mandarin "political and social convention" is a better criterion, especially when the focus of this paper is on the cognitive perspective of mental lexicon, not on the social and culture perspectives. In addition, If the authors try to propose a model for bidialectal mental lexicon, which is supposed to be different from a bilingual model, the authors will need to explain more clearly why the current results can only be accounted for by assuming that the speakers speak two dialects, instead of two languages that are rather similar and share many cognates.

According to the suggestions of the editor, Mandarin-Hakka speaker can be referred to as bilingual and bidialectal interchangeably. We agree with the suggestion because we think it is reasonable to classify bidialectal speaker as a type of bilingual. However, as far as we are concerned, we still cannot regard Mandarin and Hakka as two languages. This is because, first, Mandarin and Hakka belong to Han Chinese. They are two different variations of Han Chinese, and they both use Han writing system to symbolize the sounds. Given that Mandarin is the official language in China, it gets more developed and has Han writing system as its orthography. But as for Hakka, despite that it is a variation of Han Chinese, it does not have its own orthography although the sounds can be symbolized with Han writing system. Therefore, we regard them as two variations of Han Chinese in our paper rather than two different languages. By the way, in some research, Hakka was also regarded as a dialect (a variation of Chinese) rather than a language (e.g., Hashimoto, 1973; Lee & Zee, 2009).

As for the second suggestion, thank you very much for pointing this out. We did modify this part and add more explanation concerning this. The details can be found in the last paragraph of section 4.2.

Hashimoto, M. (1973).The Hakka Dialect: A linguistic study of its phonology syntax and lexicon. Princeton University Press.

Lee, W., & Zee, E. (2009). Hakka Chinese. Journal of the International Phonetic Association39(1), 107-111. https://doi.org/10.1017/S0025100308003599

  1. With respect to the experimental results, I still consider the small effect size to be very worrisome. Given that the model that the authors proposed is completely based on these results, from my perspective, replication experiments are indispensable.

Yes, thank you for the suggestion, and we agree with it. However, it is impossible for us to replicate the research findings within this short period of time. Thus, we listed it as a limitation of the current study in the conclusion section, and we will replicate the results with other experimental designs in the coming future.